# Potential of FDG-PET as Prognostic Significance after anti-PD-1 Antibody against Patients with Previously Treated Non-Small Cell Lung Cancer

**DOI:** 10.3390/jcm9030725

**Published:** 2020-03-07

**Authors:** Kosuke Hashimoto, Kyoichi Kaira, Ou Yamaguchi, Atsuto Mouri, Ayako Shiono, Yu Miura, Yoshitake Murayama, Kunihiko Kobayashi, Hiroshi Kagamu, Ichiei Kuji

**Affiliations:** 1Department of Respiratory Medicine, Comprehensive Cancer Center, International Medical Center, Saitama Medical University, 1397-1 Yamane, Hidaka-City, Saitama 350-1298, Japan; hkosuke@saitama-med.ac.jp (K.H.); ouyamagu@saitama-med.ac.jp (O.Y.); mouria@saitama-med.ac.jp (A.M.); respiratory@hotmail.co.jp (A.S.); you_mi@saitama-med.ac.jp (Y.M.); ymura114@saitama-med.ac.jp (Y.M.); kagamu19@saitama-med.ac.jp (H.K.); 2Department of Nuclear Medicine, Comprehensive Cancer Center, International Medical Center, Saitama Medical University, 1397-1 Yamane, Hidaka-City, Saitama 350-1298, Japan; kuji@saitama-med.ac.jp

**Keywords:** FDG-PET, immune checkpoint inhibitor, PD-1, lung cancer, prognosis

## Abstract

It remains unclear whether the accumulation of 2-deoxy-2-[^18^F]fluoro-d-glucose (^18^F-FDG) before the initiation of anti-programmed death-1 (PD-1) antibody can predict the outcome after its treatment. The aim of this study is to retrospectively examine the prognostic significance of ^18^F-FDG uptake as a predictive marker of anti-PD-1 antibody. Eighty-five patients with previously treated non-small cell lung cancer (NSCLC) who underwent ^18^F-FDG-positron emission tomography (PET) just before administration of nivolumab or pembrolizumab monotherapy were eligible in our study, and metabolic tumor volume (MTV), total lesion glycolysis (TLG) and the maximum of standardized under value (SUV_max_) on ^18^F-FDG uptake were assessed. Objective response rate, median progression-free survival and median overall survival were 36.6%, 161 days and 716 days, respectively. The frequency of any immune-related adverse events was significantly higher in patients with low ^18^F-FDG uptake on PET than in those with high uptake. By multivariate analysis, the tumor metabolic activity by TLG and MTV was identified as an independent prognostic factor for predicting outcome after anti-PD-1 antibody therapy, but not SUV_max_, predominantly in patients with adenocarcinoma. Metabolic tumor indices as TLG and MTV on ^18^F-FDG uptake could predict the prognosis after anti-PD-1 antibodies in patients with previously treated NSCLC.

## 1. Introduction

Non-small cell lung cancer (NSCLC) is the leading cause of cancer-related deaths worldwide. Recent studies have proven that anti-programmed death-1 (PD-1)/programmed death ligand-1 (PD-L1) antibodies provide significant survival benefits in patients with advanced NSCLC, when compared with standard chemotherapy [1,2,3,4]. Although the efficacy of the anti-PD-1 antibody varies according to the immunohistochemical degree of PD-L1 expression within tumor cells, there are no established biomarkers to predict the outcome after the administration of the anti-PD-1 antibody and the expression of PD-L1. If a useful biomarker is obtained from common modalities, this discovery can be easily adopted into daily practice.

Notably, 2-deoxy-2-[^18^F]fluoro-d-glucose (^18^F-FDG) positron emission tomography (PET) is a distinguished radiological modality to distinguish benign lesions from malignant tumors [5]. The uptake of ^18^F-FDG is observed in non-malignant lesions such as sarcoidosis, granuloma, pneumonia, and tuberculosis, but it can closely resemble the metabolic activity of malignant tumors [5]. Previous reports demonstrated that the accumulation of ^18^F-FDG within tumor cells was significantly linked to the presence of glucose transporter 1 (Glut1), hypoxia-inducible factor 1α (HIF-1α), and vascular endothelial growth factor (VEGF)b [6]. Several researchers have described that the accumulation of ^18^F-FDG exhibited a significant correlation with the expression of PD-L1 in patients with NSCLC [7,8,9]. Although little is known about the close association between immune environment and glucose metabolism, it is important to discover how metabolic tumor activity can affect the upregulation of PD-L1 expression. Moreover, it has been reported that ^18^F-FDG-PET is useful for assessing the therapeutic monitoring of anti-PD-1 antibody with regard to the prognosis and overall response rate [10]. Little is known as to whether the uptake of ^18^F-FDG before the administration of anti-PD-1 antibody can predict the efficacy and prognosis of immune checkpoint inhibitors (ICI) in patients with advanced NSCLC.

The maximum standardized uptake value (SUVmax) is extensively used to evaluate the metabolic degree of ^18^F-FDG within tumor cells. Since SUVmax reflects the maximal point of glucose metabolism within tumor specimens, it remains unclear whether it can indicate the total metabolic tumor volume (MTV). Recently, total lesion glycolysis (TLG) and MTV—indicators of ^18^F-FDG accumulation within tumor cells—have been identified as significant prognostic markers for predicting the treatment outcome in patients with NSCLC. A meta-analysis described TLG and MTV as better predictive markers than SUVmax [11]. Moreover, an exploratory study documented that the tumor metabolic activity assessed by TLG and MTV is better than SUVmax in evaluating the therapeutic monitoring of anti-PD-1 antibody in patients with previously treated NSCLC [10]. Recent reports suggested that baseline tumor size could predict adverse outcomes in patients with NSCLC who received ICI [12,13,14]. However, the detailed mechanism behind tumor burden (TB), determined by baseline tumor size and causing poor efficacy of ICI, is unclear. Tumors possibly suppress the immune response by different mechanisms other than the PD-1/PD-L1 pathway. Tumor hypoxia, determined by HIF-1, induces VEGF, which induces immunosuppressive T-lymphocytes such as regulatory T-cells (Tregs) and myeloid-derived suppressor cells [15,16]. These evidences suggest that the high HIF-1 expression caused by an increased tumor size creates an immunosuppressive environment regardless of ICI treatment and contributes to the poor outcome for patients with NSCLC [15,16]. However, whether morphological assessment, based on baseline tumor size, can accurately reflect the volume of tumor hypoxia or tumor metabolic activity is debatable. Therefore, ^18^F-FDG uptake is expected to assess tumor activity more accurately than TB on computed tomography (CT).

We conducted this study to investigate whether the degree of ^18^F-FDG uptake before the administration of anti-PD-1 antibody can predict the prognosis in patients with previously treated advanced NSCLC, by evaluating the correlation between TB and SUVmax, TLG, and MTV.

## 2. Materials and methods

### 2.1. Patients

We retrospectively examined the medical records at Saitama Medical University International Medical Center, Saitama, Japan (ethical approval code: 19-225; date of approval: 13 November 2019) and selected patients with previously treated NSCLC who received anti-PD-1 antibody monotherapy, such as nivolumab and pembrolizumab, and underwent ^18^F-FDG-PET after previous treatment and before the initiation of anti-PD-1 antibody as a recurrent survey. From February 2016 through April 2019, 97 patients with pretreated NSCLC were administered nivolumab and pembrolizumab. Twelve patients were excluded because of inadequate medical information and absence of an evaluable target lesion. Therefore, the final cohort consisted of 85 patients. 

This study was approved by the institutional ethics committee of the Saitama Medical University International Medical Center. All procedures performed in studies involving human participants were in accordance with the ethical standards of the institutional and/or national research committee and with the 1964 Helsinki declaration and its later amendments or comparable ethical standards. The requirement for written informed consent was waived because of the retrospective nature of the study.

### 2.2. Treatment, Efficacy Evaluation, and Assessment of Baseline Tumor Burden

Nivolumab and pembrolizumab were intravenously administered at 3 mg/kg every 2 weeks and at 200 mg/day every 3 weeks, respectively. Complete blood cell count, differential count, routine chemistry measurements, physical examination, and toxicity assessment were performed weekly. Acute toxicity was graded using the Common Terminology Criteria for Adverse Events version 4.0. Tumor response was evaluated using the Response Evaluation Criteria in Solid Tumors version 1.1 [17]. Baseline TB was evaluated using CT for target lesions [13,14]. TB was defined as the sum of the longest diameter for a maximum of five target lesions and up to two lesions per organ [13,14].

### 2.3. PET Imaging and Data Analysis

Patients fasted for at least 6 hours before PET imaging, performed using a PET/CT scanner (Biograph 6 or 16, Siemens Healthineers K.K., Japan) with a 585 mm field of view. Three-dimensional data acquisition was initiated 60 minutes after injecting 3.7 MBq/kg of FDG. We acquired eight bed positions (2-minute acquisition per bed position) according to the range of imaging. Attenuation-corrected transverse images obtained with ^18^F-FDG were reconstructed with the ordered-subsets expectation-maximization algorithm, based on the point spread function into 168 × 168 matrices with a slice thickness of 2.00 mm.

For the semiquantitative analysis, functional images of SUV were produced using attenuation-corrected transaxial images, injected dosage of ^18^F-FDG, patient’s body weight, and the cross-calibration factor between PET and the dose calibrator. SUV was defined as follows:

SUV = Radioactive concentration in the region of interest (ROI) (MBq/g)/Injected dose (MBq)/Patient’s body weight (g).

A nuclear physician conducted the volume of interest (VOI) analysis using CT scans, eliminating the physiological uptake in the heart, urinary tracts, and gastrointestinal tracts. We used GI-PET software (Nihon Medi-physics Co. Ltd., Tokyo, Japan) on a Windows workstation to semi-automatically calculate the MTV and TLG (= SUV_mean_ × MTV), of each lesion using SUV thresholds in the liver VOI (= SUV_mean_ + [1.5 × SUV_Standard_Deviation_]). These SUV thresholds were the optimum values to generate VOIs in which the whole tumor mass is completely enclosed in all cases, with the CT image as the reference. SUV_max_ and SUV_mean_ within the generated VOI were also calculated automatically. VOIs over all measurable lesions on pretreatment PET/CT were automatically registered. In case of multiple lesions in the same organ, a maximum of 100 lesions were measured.

### 2.4. Statistical Analysis

Statistical significance was indicated by *p* < 0.05. Fisher’s exact tests were used to examine the association between two categorical variables. Correlations between SUVmax, MTV, and TLG on ^18^F-FDG uptake were analyzed using the Pearson rank test. The Kaplan–Meier method was used to estimate survival as a function of time, and survival differences were analyzed by log-rank tests. Progression-free survival (PFS) was defined as the time from the initiation of anti-PD-1 antibody to tumor recurrence or death from any cause, while overall survival (OS) was defined as the time from the initiation of anti-PD-1 antibody to death from any cause. Statistical analyses were performed using GraphPad Prism 7 (Graph Pad Software, San Diego, CA, USA) and JMP 14.0 (SAS Institute Inc., Cary, NC, USA).

## 3. Results

### 3.1. Assessment of SUV_max_, MTV, and TLG on ^18^F-FDG Uptake

In all 85 patients, the median values of SUVmax, MTV, and TLG on ^18^F-FDG and TB were 6.0 (range, 3.1–21.0), 17.8 cm^3^ (range, 1.1–379 cm^3^), 75.4 gcm^3^/mL (range, 3.9–2550 g·cm^3^/mL), and 65 cm (range, 6.6–230.6 cm), respectively. According to histological types, the median values of SUVmax, MTV, and TLG on ^18^F-FDG and TB in adenocarcinoma were 6.7 (range, 3.1–21.3), 11.3 cm^3^ (range, 1.1–276 cm^3^), 58.3 g·cm^3^/mL (range, 3.9–1398 g·cm^3^/mL), and 53 cm (range, 6.6–230.6 cm), respectively, and those in non-adenocarcinoma displayed 8.4 (range, 3.2–20.4) 24.8 cm^3^ (range, 1.1–379 cm^3^), 105.6 g·cm^3^/mL (range, 4.0–2550 g·cm^3^/mL), and 75 cm (range, 18.2–228.3 cm), respectively. The difference of each parameter for SUVmax, MTV, and TLG on ^18^F-FDG and TB was not significant between adenocarcinoma and non- adenocarcinoma. The SUVmax correlated significantly with MTV (*r* = 0.49, *p* < 0.01), TLG (*r* = 0.56, *p* < 0.01), and TB (*r* = 0.33, *p* < 0.01). Figure 1 is a representative PET image showing the assessment of SUVmax, MTV, and TLG on ^18^F-FDG.

The discriminative value of various SUVmax, MTV, and TLG cutoffs for ^18^F-FDG uptake were explored in the context of OS and PFS (Figure 2) [13]. For prognosis in OS and PFS analyses, the most discriminative cutoffs based on log-rank test for SUVmax, MTV, and TLG were 6.0, 5.0, and 20, respectively. The TLG cutoff of 20 was significant in the OS analysis, but not in the PFS analysis, although it was the most favorable TLG cutoff. The SUVmax cutoff of 6 was not significant in either OS or PFS analysis, but still seemed better considering the results of the log-rank test. The 12 cm cutoff for TB was based on a previous study [13]. 

### 3.2. Patient Demographics

Patient demographics according to the cutoff values of SUVmax, MTV, and TLG on ^18^F-FDG uptake are listed in Table 1. In the 17 patients harboring epidermal growth factor receptor (*EGFR)* mutation, deletion 19 and L858R were observed in 11 and 6 patients, respectively. High TLG and SUVmax on ^18^F-FDG uptake were significantly associated with smoking history. The objective response rate and disease control rate were 36.6% [95% confidence internal (CI); 26.2%–47.0%] and 65.9% [95%CI; 55.6%–76.1%]. No significant difference in the response to ICI was observed according to the degree of SUVmax, MTV, and TLG on ^18^F-FDG uptake. The median time from the date of ^18^F-FDG-PET scan to the initiation of anti-PD-1 antibodies was 18 days (range, 1–107 days). Next, we analyzed different incidences of immune-related adverse events (irAEs) according to the degree of SUVmax, MTV, and TLG on ^18^F-FDG uptake and TB. The frequency of any irAE was significantly higher in patients with low values of SUVmax, MTV, and TLG on PET than in those with high values, but not for TB (Appendix A, online only). The incidence of grade 3 or 4 irAEs exhibited no close correlation with the degree of ^18^F-FDG uptake on PET and TB.

### 3.3. Survival and ^18^F-FDG-PET:

The median OS and PFS were 716 and 161 days, respectively, and the 2-year OS rate was 44.8%. Among all patients, those with a low TLG exhibited significantly better OS than those with a high TLG. Similarly, the survival analysis of MTV on ^18^F-FDG uptake demonstrated a significantly worse OS and PFS for patients with a high MTV (Figure 2). No statistically significant difference in the OS and PFS was observed between patients with a low and high SUVmax (Figure 2). However, in 70 patients without an EGFR mutation, a statistically significant difference in OS and PFS was recognized between patients with low and high TLG or low and high MTV, but not between those with low and high SUVmax and low and high TB (Figure 3). Survival analysis results are listed in Table 2. Univariate analysis in all patients identified performance status (PS), TLG, and MTV as significant prognostic markers for OS; the significant predictors for PFS were PS and MTV. Subsequently, we performed a multivariate analysis according to TLG and MTV and confirmed that PS, TLG, and MTV were independent prognostic factors for poor OS and PFS; the significant prognostic marker for PFS was PS (Table 3). In patients without EGFR mutation, a multivariate analysis identified TLG and MTV as independent prognostic factors for predicting poor OS. Histological typing confirmed TLG and MTV as significant prognostic factors in patients with adenocarcinoma, but not in those with non-adenocarcinoma.

## 4. Discussion

To the best of our knowledge, this is the first study to evaluate the prognostic significance of metabolic parameters measured by ^18^F-FDG-PET for predicting outcomes after the initiation of anti-PD-1 antibodies in patients with previously treated NSCLC. We found that the tumor metabolic volume assessed by TLG and MTV was an independent prognostic factor, but not SUVmax and TB. The roles of TLG and MTV as significant predictive markers after anti-PD-1 antibody therapy may be closely associated with patients with a histology of adenocarcinoma and without EGFR mutation. Although the value of SUVmax on ^18^F-FDG uptake within tumor cells was closely correlated with that of TLG and MTV, SUVmax on ^18^F-FDG uptake before anti-PD-1 antibody could not accurately predict the outcome after the initiation of treatment. Furthermore, we found that low uptake of ^18^F-FDG was closely associated with the occurrence of irAEs and the incidence of grade 3 or 4 irAEs exhibited no close correlation with the degree of ^18^F-FDG uptake on PET. In addition, there was no significant correlation between the response to the anti-PD-1 antibody and the degree of ^18^F-FDG uptake. However, we discovered that MTV of pretreatment ^18^F-FDG-PET plays a crucial role in the prognostic significance of anti-PD-1 antibody treatment. Unlike previous studies [12,13], our study indicated that TB determined by baseline tumor size was not a significant predictor of ICI or irAEs. Further studies are warranted to investigate whether pretreatment of MTV can predict the outcome of ICI plus ICI or the combination of ICI plus cytotoxic chemotherapy as first-line treatment.

A recent study reported that the metabolic response to ^18^F-FDG uptake measured by TLG or MTV was a stronger biomarker for the prediction of efficacy and survival at one month after the initiation of nivolumab than SUV_max_ [10]. The predictive probability of partial response (100% versus 29%, *p* = 0.021) and progressive disease (100% versus 22.2%, *p* = 0.002) at one month after nivolumab treatment was significantly higher in ^18^F-FDG-PET than in CT. Moreover, multivariate analysis confirmed the ^18^F-FDG uptake measurement by TLG or MTV as an independent prognostic factor. Thus, ^18^F-FDG-PET may be a useful radiographic modality for immune monitoring to predict the efficacy of ICI. However, a large-scale prospective study is required to confirm these results [10,18]. A recent meta-analysis suggested that in lung cancer, metabolic parameters such as TLG and MTV are better prognostic predictors after any treatment when compared with SUV_max_ [11].

Recently, Jreige et al. described that the metabolic-to-morphological volume ratio (MMVR)—calculated by dividing MTV by morphological tumor volume (MoTV)—was able to predict the radiological response to anti-PD-1/PD-L1 blockade and was negatively correlated with tumor PD-L1 expression and tumor necrosis [18]. Although their study analyzed only 17 patients who received anti-PD-1 therapy, they suggested that anti-PD-1 therapy is effective for tumor tissues with a low ratio of MTV, and the presence of a low MTV contributes to the upregulation of PD-L1 expression by the hypoxic environment in the tumor arising from necrosis and inflammation. However, it remained unclear whether MMVR could predict the outcome after anti-PD-1 therapy. Our study, however, indicated that patients with a low MTV exhibited a favorable prognosis after anti-PD-1 therapy. Several studies have demonstrated that tumor necrosis triggers inflammation, which facilitates the influx of lymphocytes and the upregulation of PD-L1, related to tumor necrosis factor-α, and that necrosis or inflammation was closely linked to tumor recurrence and worse survival [19,20]. Our study could not investigate the relationship between necrosis or inflammation and tumor metabolic volume; therefore, it remains unclear whether low MTV and TLG are associated with necrosis, inflammation, and hypoxic environment in the tumor. Further investigation is warranted to elucidate the prognostic significance of MTV and TLG as predictive markers of anti-PD-1 therapy based on the biological aspect.

The mechanism by which tumor cells uptake ^18^F-FDG requires glucose metabolism, hypoxia, and angiogenesis, and ^18^F-FDG is closely associated with the expression of these markers [6]. The expression of PD-L1 is significantly related to Glut1, HIF-1α and SUV_max_ on ^18^F-FDG uptake in lung cancer [7,8,9]. However, it is reported that TLG and MTV on ^18^F-FDG uptake were not closely correlated with PD-L1 expression levels [18], but Foxp3-Tregs are described to be positively associated with TLG and MTV [21]. Considering these evidences, the active environment with high MTV may form an immunosuppressive state, contributing to the resistant situation to ant-PD-1 blockage.

Our study has several limitations. First, we employed a retrospective approach and a small cohort, which may have introduced bias in our results. More than half of the patients with previously treated NSCLC had not undergone ^18^F-FDG-PET just before the initiation of anti-PD-1 antibody treatment. Therefore, we think that the number of eligible patients was slightly limited. However, the current study is meaningful for the prediction of anti-PD-1 therapy. Second, our study suggests that MTV could be a predictive marker for ICI treatment compared to the morphological tumor volume. However, the appropriateness of morphological assessment used for calculating the optimal tumor volume of all metastatic lesions using CT scan is debatable. Thus, it may be difficult to perform a concise comparison between metabolic and morphological tumor volumes. Moreover, we had no information about the calculation of MMVR in our study, thus, there were some limitations in discussing the prognostic significance of metabolic tumor volume compared to MMVR. Lastly, an the immunohistochemical analysis of PD-L1, expression within tumor cells using tumor specimens was not performed before anti-PD1 antibody treatment. At our institution, re-biopsy was not done routinely before ICI treatment except for the initial diagnosis. Further study is warranted to investigate the relationship between MTV and the immune environment.

In conclusion, TLG and MTV on ^18^F-FDG uptake may predict the prognosis after anti-PD-1 antibodies in patients with previously treated NSCLC, but not SUVmax on ^18^F-FDG uptake. Although MTV may indirectly reflect the presence of morphological tumor volume with active tumor cells, the assessment of TLG and MTV on ^18^F-FDG uptake is easily executable in daily practice. A prospective study is needed to confirm these results.

## Figures and Tables

**Figure 1 jcm-09-00725-f001:**
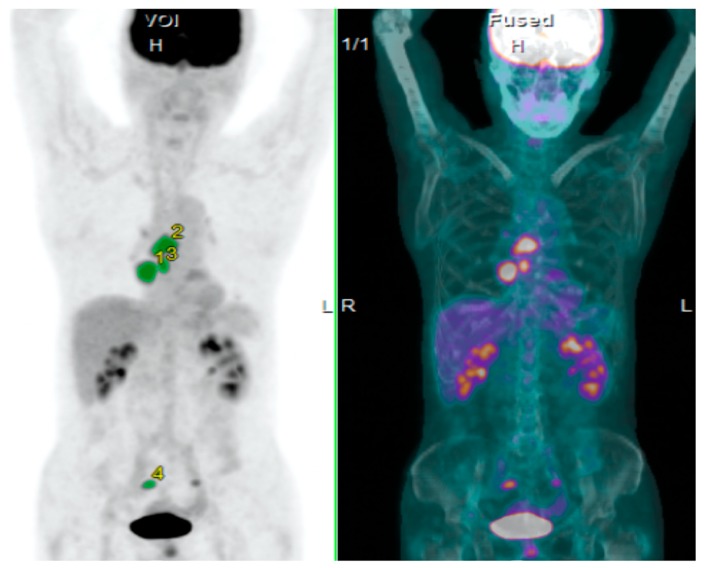
Imaging from a positron emission tomography (PET) scan of non-small cell lung cancer (NSCLC), indicating the measurement of SUVmax, metabolic tumor volume (MTV) and total lesion glycolysis (TLG) on 2-deoxy-2-[^18^F]fluoro-d-glucose (^18^F-FDG).

**Figure 2 jcm-09-00725-f002:**
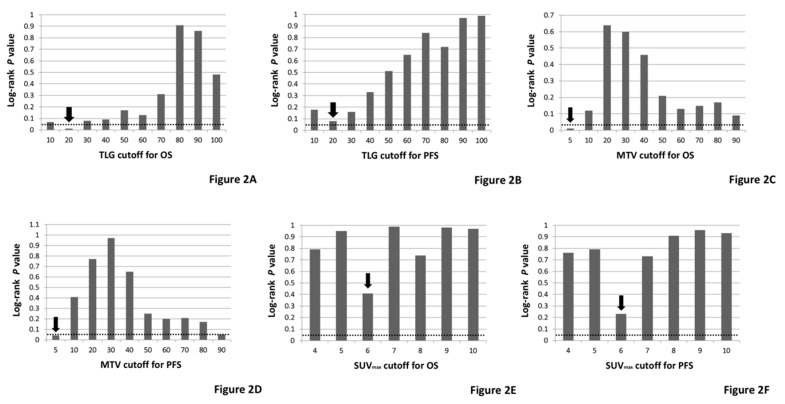
Discriminative value by log-rank test according to various TLG (**A**), MTV (**C**) and SUVmax (**E**) cutoff for OS and TLG (**B**), MTV (**D**) and SUVmax (**F**) cutoff for PFS in 18F-FDG-PET.

**Figure 3 jcm-09-00725-f003:**
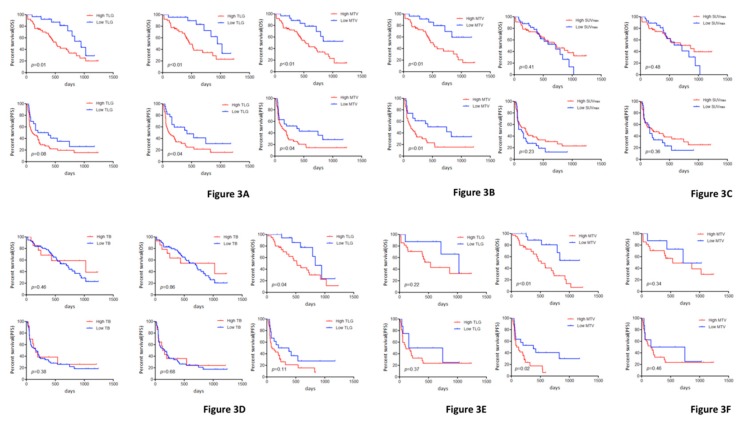
Kaplan–Meier curves according to various TLG (**A**), MTV (**B**), SUVmax (**C**) and TB (**D**) for OS and PFS. (**A**); OS (upper left) and PFS (lower left) in all patients and OS (upper right) and PFS (lower right) in those without EGFR mutation according to TLG. (**B**); OS (upper left) and PFS (lower left) in all patients and OS (upper right) and PFS (lower right) in those without EGFR mutation according to MTV. (**C**); OS (upper left) and PFS (lower left) in all patients and OS (upper right) and PFS (lower right) in those without EGFR mutation according to SUVmax. (**D**); OS (upper left) and PFS (lower left) in all patients and OS (upper right) and PFS (lower right) in those without EGFR mutation according to TB. (**E**); OS (upper left) and PFS (lower left) in patients with AC patients and OS (upper right) and PFS (lower right) in those with non-AC according to TLG. (**F**); OS (upper left) and PFS (lower left) in patients with non-AC patients and OS (upper right) and PFS (lower right) in those with non-AC according to MTV. Abbreviation: TLG, total lesion glycolysis; MTV, metabolic tumor volume; SUVmax: maximal standardized under value; TB, tumor burden; OS, overall survival; PFS, progression-free survival; AC, adenocarcinoma; non-AC, non-adenocarcinoma.

**Table 1 jcm-09-00725-t001:** Patient’s demographics according to the assessment of FDG uptake.

Variables	Total (*n* = 85)	TLG			MTV			SUVmax		
		High (*n* = 59)	Low (*n* = 26)	*p*-Value	High (*n* = 58)	Low (*n* = 27)	*p*-Value	High (*n* =5 2)	Low (*n* = 33)	*p*-Value
**Age** **≤ 70/>70** **(range: 38–86 years)**	42/43	29/30	13/13	>0.99	25/33	17/10	0.11	26/26	16/17	>099
**Gender** Male/Female	65/20	47/12	18/8	0.41	46/12	19/8	0.42	42/10	23/10	0.29
**Smoking history** Yes/No	71/14	51/8	10/16	**<0.01**	50/8	21/6	0.35	47/5	24/9	**0.04**
**Performance status** 0 or 1/2 or 3	79/6	53/6	26/0	0.17	52/6	27/0	0.17	46/6	33/0	0.07
**Histological type** AC/Non-AC	51/34	33/26	18/8	0.34	32/26	19/8	0.24	27/25	24/9	0.07
**EGFR mutation** Yes/No	17/68	12/47	5/21	>0.99	12/47	5/22	>0.99	10/42	7/26	>0.99
**Response to ICI ^#^** CR or PR/SD or PD CR, PR or SD/PD	29/51 53/27	20/35 34/21	9/16 19/6	>0.99 0.31	20/34 36/18	9/17 17/9	>0.99 >0.99	21/27 34/15	8/24 19/12	0.10 0.47

Abbreviations: TLG, total lesion glycolysis; MTV, metabolic tumor volume; SUVmax, the maximum of standardized uptake value; AC, adenocarcinoma; Non-AC, non-adenocarcinoma; EGFR, epidermal growth factor receptor; CR, complete response; PR, partial response; SD, stable disease; PD, progressive disease; ICI, immune checkpoint inhibitor. #, because of 6 patients with no measurable lesion, 82 patients were analyzed according to the uptake of FDG; Bold character shows statistically significance.

**Table 2 jcm-09-00725-t002:** Univariate analysis in overall survival and progression-free survival.

Variables	Overall Survival				Progression-Free Survival			
	All Patients (*n* = 85)		Patients without EGFR Mutation (*n* = 70)		All Patients (*n* = 85)		Patients without EGFR Mutation (*n* = 70)	
	MST (days)	*p*-Value	MST (days)	*p*-Value	MST (days)	*p*-Value	MST (days)	*p*-Value
**Age** (≤70/>70)	737/693	0.73	865/693	0.85	164/161	0.88	181/180	0.92
**Gender** (Male/Female)	716/737	0.85	693/837	0.59	181/75	0.67	172/420	0.23
**Smoking** (Yes/No)	716/737	0.53	716/637	0.74	181/72	0.09	181/272	0.85
**PS** (0 or 1/2 or 3)	724/115	**<0.01**	716/74	**<0.01**	172/40	**<0.01**	200/25	0.10
**Histological type** (AC/Non-AC)	724/716	0.84	693/716	0.66	146/161	0.76	220/161	0.39
**TLG** (High/Low)	516/945	**0.01**	465/945	**<0.01**	114/291	0.08	114/420	**0.04**
**MTV** (High/Low)	536/NR	**<0.01**	465/NR	**<0.01**	125/382	**0.04**	127/382	**0.02**
**SUVmax** (High/Low)	724/716	0.41	865/716	0.48	201/125	0.23	204/161	0.36
**TB** (High/Low)	793/693	0.46	837/693	0.86	204/137	0.38	182/181	0.68

Abbreviations: TLG, total lesion glycolysis; MTV, metabolic tumor volume; SUVmax, the maximum of standardized uptake value; AC, adenocarcinoma; Non-AC, non-adenocarcinoma; PS, performance status; HR, hazard ratio; 95% CI, 95% confidence interval; NR, not reached; EGFR, epidermal growth factor receptor; TB, tumor burden; Bold character shows statistically significance.

**Table 3 jcm-09-00725-t003:** Multivariate analysis of overall survival and progression-free survival.

Variables	Overall Survival				Progression-Free Survival			
	All Patients (*n* = 85)		Patients without EGFR Mutation (*n* = 70)		All Patients (*n* = 85)		Patients without EGFR Mutation (*n* = 70)	
	HR 95% CI	*p*-Value	HR 95% CI	*p*-Value	HR 95% CI	*p*-Value	HR 95% CI	*p*-Value
**Survival Analysis Including TLG**								
**Age** (≤ 70/> 70)	1.02 0.74–1.39	0.91	1.02 0.72–1.44	0.90	0.95 0.74–1.23	0.74	0.98 0.73–1.31	0.89
**Gender** (Male/Female)	1.11 0.76–1.55	0.55	0.95 0.55–1.48	0.85	1.07 0.79–1.41	0.61	0.81 0.49–1.19	0.31
**PS** (0 or 1/2 or 3)	1.68 1.05–2.51	**0.03**	2.22 1.06–3.93	**0.03**	1.63 1.02–2.36	**0.04**	1.41 0.68–2.43	0.31
**TLG** (High/Low)	1.47 1.03–2.21	**0.03**	1.63 1.10–2.60	**0.01**	1.21 0.92–1.63	0.16	1.32 0.97–1.86	0.07
**Survival Analysis Including MTV**								
**Age** (≤ 70/> 70)	0.92 0.68–1.26	0.64	0.87 0.61–1.25	0.46	0.91 0.71–1.18	0.51	0.91 0.67–1.21	0.51
**Gender** (Male/Female)	1.09 0.75–1.52	0.61	0.97 0.56–1.51	0.91	1.06 0.78–1.40	0.66	0.80 0.49–1.18	0.28
**PS** (0 or 1/2 or 3)	1.69 1.06–2.51	**0.02**	2.23 1.06–3.94	**0.03**	1.59 1.02–2.33	**0.04**	1.41 0.68–2.41	0.31
**MTV** (High/Low)	1.59 1.09–2.45	**0.01**	1.83 1.19–3.04	**<0.01**	1.28 0.97–1.73	0.07	1.45 1.05–2.05	**0.02**

Abbreviations: TLG, total lesion glycolysis; MTV, metabolic tumor volume; PS, performance status; HR, hazard ratio; 95% CI, 95% confidence interval; EGFR, epidermal growth factor receptor.

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
