# Peer review of "Potential of FDG-PET as Prognostic Significance after anti-PD-1 Antibody against Patients with Previously Treated Non-Small Cell Lung Cancer"

_jcm, 2020, doi:10.3390/jcm9030725_

Round 1
Reviewer 1 Report
- It is a nice study, well conducted, with an interesting purpose. The authors recognize their methodological limitation, but the study provide more data for prognosis before PD-1 antibody against treatment in NSCLC.
- I kindly ask to the authors about some points:
- Did they found any difference between nivolumab and pembrolizumab?
- Can be some bias due to the previous applied treatments?
- How the authors explain the relationship between pre-treatment MTV and the successful anti PD-1 antibody treatment? Why the degree of FDG uptake do not have influence on this issue?
- Can be better to consider FDG-PET/CT as a molecular imaging technique or a nuclear medicine technique than a radiological technique.
Author Response
Comments to reviewer 1
- It is a nice study, well conducted, with an interesting purpose. The authors recognize their methodological limitation, but the study provide more data for prognosis before PD-1 antibody against treatment in NSCLC.
- I kindly ask to the authors about some points:
Re) Thank you for generous comments.
- Did they found any difference between nivolumab and pembrolizumab?
Re) nivolumab and pembrolizumab are anti-PD-1 antibody, and there is no significant difference in the pharmacological action both drugs.
- Can be some bias due to the previous applied treatments?
Re) Several kinds of chemotherapy were performed before anti-PD-1 antibodies, however, we consider that the previous treatment didn’t bias the results of our study.
- How the authors explain the relationship between pre-treatment MTV and the successful anti PD-1 antibody treatment? Why the degree of FDG uptake do not have influence on this issue?
Re) Since metabolic tumor activity by MTV is considered to concisely reflect the tumor volume, the high HIF-1 expression caused by an increased tumor size creates an immunosuppressive environment and Foxp3-Tregs are described to be positively associated with MTV, we think that the increased MTV forms a negative tumor immune microenvironment and is resistant to PD-1 blockade regardless of an increased PD-L1 expression.
- Can be better to consider FDG-PET/CT as a molecular imaging technique or a nuclear medicine technique than a radiological technique.
Re) The present study showed that metabolic tumor activity by MTV is better to predict the outcome after anti-PD-1 antibody than baseline tumor size. To evaluate the metabolic tumor activity, molecular imaging technique such FDG-PET is more suitable as compared with radiological technique.
Reviewer 2 Report
The Authors investigate on 85 patients whetever the degree of 18F-FDG uptake before the administration of anti-PD-1 antibody can predict the prognosis in patients with previously treated advance NSCLC by evaluate the correlation between TB and SUVmax, TLG, and MTV.
The Authors found the tumor metabolic volume assed by TLG and MTV was an independent prognostic factor for predicting outcome after anti-PD1 antibody therapy only on adenocarcinoma.
Minor revision:
1) Table A1 Please add the indication of gender.
2) Table 1. The Authours should include the age distribution for discriminate the effect of the age on the uptake.
3) The Authors showed the low uptake was associated with irAEs. The Authors should comment in discussion part the fact the grade 3 and 4 are not significant associated .
4) The Authors should describe the mutation state of all patience if they now the status and the type of the mutation. If it is not possible Authours should include this as a limitation of the work in the discussion section.
5) The Authors on row 197 comment the study of Jreiege et al. Moreover, these discussion are not completely explained from the data presented in this paper. The Authors should include this calculation of MMVR for a better association. If it is not possible Authours should include this as a limitation of the work in the discussion section.
Author Response
Comments to reviewer 2
The Authors investigate on 85 patients whetever the degree of 18F-FDG uptake before the administration of anti-PD-1 antibody can predict the prognosis in patients with previously treated advance NSCLC by evaluate the correlation between TB and SUVmax, TLG, and MTV.
The Authors found the tumor metabolic volume assed by TLG and MTV was an independent prognostic factor for predicting outcome after anti-PD1 antibody therapy only on adenocarcinoma.
Re) Thank you for your generous comments.
Minor revision:
1) Table A1 Please add the indication of gender.
Re) According to reviewer’s suggestion, the indication of gender was inserted in Table A1.
2) Table 1. The Authours should include the age distribution for discriminate the effect of the age on the uptake.
Re) According to reviewer’s suggestion, the age distribution was added in Table 1.
3) The Authors showed the low uptake was associated with irAEs. The Authors should comment in discussion part the fact the grade 3 and 4 are not significant associated.
Re) According to reviewer’s suggestion, we commented in the discussion section. The sentence of “and the incidence of grade 3 or 4 irAEs exhibited no close correlation with the degree of 18F-FDG uptake on PET” was added in discussion.
4) The Authors should describe the mutation state of all patience if they now the status and the type of the mutation. If it is not possible Authours should include this as a limitation of the work in the discussion section.
Re) According to reviewer’s suggestion, the mutation status of EGFR was described in the results section (3.2 patient’s demographics). In the 17 patients harboring EGFR mutation, deletion 19 and L858R were observed in 11 and 6 patients, respectively.
5) The Authors on row 197 comment the study of Jreiege et al. Moreover, these discussion are not completely explained from the data presented in this paper. The Authors should include this calculation of MMVR for a better association. If it is not possible Authours should include this as a limitation of the work in the discussion section.
Re) Thank you for your generous comments. In the discussion section, the sentence of “Moreover, we had no information about the calculation of MMVR in our study, thus, there was some limitation to discuss the prognostic significance of metabolic tumor volume compared to MMVR” was added as study limitation.